# Is Photoprotection of PSII One of the Key Mechanisms for Drought Tolerance in Maize?

**DOI:** 10.3390/ijms222413490

**Published:** 2021-12-16

**Authors:** Nahidah Bashir, Habib-ur-Rehman Athar, Hazem M. Kalaji, Jacek Wróbel, Seema Mahmood, Zafar Ullah Zafar, Muhammad Ashraf

**Affiliations:** 1Department of Botany, The Women University, Multan 66000, Pakistan; 2Institute of Pure and Applied Biology, Bahauddin Zakariya University, Multan 60800, Pakistan; habibathar@yahoo.com (H.-u.-R.A.); drseemapk@gmail.com (S.M.); zafarbzu@yahoo.com (Z.U.Z.); 3Department of Plant Physiology, Institute of Biology, University of Life Sciences SGGW, 02-776 Warsaw, Poland; 4Institute of Technology and Life Sciences, National Research Institute, Falenty, Al. Hrabska 3, 05-090 Raszyn, Poland; 5Department of Bioengineering, West Pomeranian University of Technology in Szczecin, Słowackiego 17, 71-434 Szczecin, Poland; Jacek.Wrobel@zut.edu.pl; 6Institute of Molecular Biology and Biotechnology, The University of Lahore, Lahore 54590, Pakistan; ashrafbot@yahoo.com

**Keywords:** nonphotochemical quenching, cyclic electron transport, donor-end limitations to PSI, yield, 100-kernal weight

## Abstract

Drought is one of the most important abiotic stress factors limiting maize production worldwide. The objective of this study was to investigate whether photoprotection of PSII was associated with the degree of drought tolerance and yield in three maize hybrids (30Y87, 31R88, P3939). To do this, three maize hybrids were subjected to three cycles of drought, and we measured the activities of photosystem II (PSII) and photosystem I (PSI). In a second field experiment, three maize hybrids were subjected to drought by withholding irrigation, and plant water status, yield and yield attributes were measured. Drought stress decreased leaf water potential (Ψ_L_) in three maize hybrids, and this reduction was more pronounced in hybrid P3939 (−40%) compared to that of 30Y87 (−30%). Yield and yield attributes of three maize hybrids were adversely affected by drought. The number of kernels and 100-kernel weight was the highest in maize hybrid 30Y87 (−56%, −6%), whereas these were lowest in hybrid P3939 (−88%, −23%). Drought stress reduced the quantum yield of PSII [Y(II)], photochemical quenching (qP), electron transport rate through PSII [ETR(II)] and NPQ, except in P3939. Among the components of NPQ, drought increased the Y(NPQ) with concomitant decrease in Y(NO) only in P3939, whereas Y(NO) increased in drought-stressed plants of hybrid 30Y87 and 31R88. However, an increase in cyclic electron flow (CEF) around PSI and Y(NPQ) in P3939 might have protected the photosynthetic machinery but it did not translate in yield. However, drought-stressed plants of 30Y87 might have sufficiently downregulated PSII to match the energy consumption in downstream biochemical processes. Thus, changes in PSII and PSI activity and development of NPQ through CEF are physiological mechanisms to protect the photosynthetic apparatus, but an appropriate balance between these physiological processes is required, without which plant productivity may decline.

## 1. Introduction

Increasing sustainable crop productivity under water-limited conditions by developing drought-tolerant cultivars is one of the most challenging issues for the plant scientist [1,2,3]. Drought-tolerant plants or drought-tolerant cultivars of same species had a better ability to maintain physiological activities such as CO_2_ assimilation rate, nutrient uptake, etc., at low-water status [4]. Since plant growth and yield mainly depend on plant photosynthetic activity, several researchers are of the view that selection of cultivars/varieties based on photosynthetic traits may help in developing high-yielding and stress-tolerant crop cultivars [5,6,7,8,9,10]. Most of the studies have focused on improving the activity of rubisco with enhanced CO_2_ fixation, or lowering carbon loss through photorespiration [7,8,9,10]. Similarly, some are of view that counteracting drought stress induces ROS production, and improving solar energy capture by photosystems may enhance the crop yield [11,12,13]. However, only a 10–15% crop improvement using some photosynthetic traits is possible, and this is mainly due to poor understanding about plant responses to drought stress [10].

Drought stress inhibits the rate of photosynthesis by reducing stomatal conductance and influx of CO_2_ [14,15], resulting in an imbalance in generation and utilization of ATP in the Calvin cycle. This situation causes over-reduction of electron carriers (at photosystem II (PSII), cytb_6_f complex and photosystem I (PSI)), and accumulation of reactive oxygen species (ROS). Over-reduction of electron carriers of electron transport resulted in photoinhibition and/or photodamage of photosystem II (PSII) [16]. Similarly, over-reduction of the acceptor side of photosystem I resulted in over-build-up of NADPH and ROS generation, thereby resulting in photoinhibition of photosystem I PS(I) under drought [17].

During electron transport from PSII to PSI, ROS are produced by photoreduction of O_2_ via the FeS centers at the acceptor side of photosystem I (Mehler reaction) [18], and singlet oxygen (^1^O_2_) species can be generated when molecular oxygen interacts with triplet-excited chlorophyll. The former occurs when electron acceptors of PSI (particularly centers of FeS and ferredoxin) are reduced, while the latter is associated with overexcitation of PSII under abiotic stress conditions including drought. Photosystem I is more protected than PSII by a battery of antioxidants in the stroma, i.e., superoxide dismutase (SOD), peroxidase (POD) and catalase (CAT) [11,13]. Photodamage of photosystem II (PSII) is thus considered as a harmful physiological response that decreases plant productivity under different environmental stresses. Development of nonphotochemical quenching (NPQ) and cyclic electron flow safely dissipate excessive energy absorbed by PSII as heat, to protect PSII from photodamage [19,20,21,22]. However, controlled oxidative damage of PSII or D1 protein of PSII, i.e., dynamic photoinhibition, is another potential strategy to protect the rest of photosynthetic machinery, as has been observed in lotus plants [23].

Photosynthetic acclimatory responses in plants against drought stress include development of nonphotochemical quenching (NPQ) and cyclic electron flow [19,20,21,22]. However, NPQ develops due to the generation of a pH gradient across the thylakoid membrane via linear electron flow (LEF) and cyclic electron flow (CEF) [24]. Drought reduces electron flux through the LEF route and increases through CEF [25]. Moreover, CEF generates the ΔpH to stimulate NPQ, and consequently protects the PS(II). Greater stimulation of CEF and NPQ has been observed in plants of drought-tolerant *Jatropha curcas* when compared to in drought-sensitive plants of *Ricinus communis* [21].

In view of the above-mentioned studies, it is stated that the capability of stress-responsive capability regarding oxidative damage and changes in activity of photosynthetic machinery are key physiological characteristics which manifest drought-tolerance in plants. It is well-established that measurement of the efficiency of PSII, or extent of photoinhibition using a chlorophyll fluorescence meter, is generally associated with plant health or stress tolerance and can be used as potential physiological selection criteria for drought tolerance. Thus, the key objective of this research was to examine photosynthetic acclimatory responses in three maize hybrids differing in drought tolerance. The secondary objective of the study was to assess whether or not the degree of drought tolerance in these maize hybrids was associated with photoprotection of PSII.

## 2. Results

### 2.1. Light-Response Curve

#### 2.1.1. Measurement of Activity of PSII, ETR(II), qP

Quantum yield of PSII [Y(II)] decreased significantly as a function of light intensity (11 to 830 μmol m−2 s−1) in all three maize hybrids (Table 1; Figure 1). Drought stress also reduced the Y(II) in all three maize hybrids. Similarly, photochemical quenching (qP) also decreased as a function of light intensity in all the three maize hybrids (Figure 2). Drought stress significantly reduced both Y(II) and qP in all three maize hybrids at 100–500 μmol m−2 s−1 PAR (Table 2; Figure 1). However, drought-induced reduction in Y(II) and qP was more pronounced in hybrids 30Y87 and 31R88. Moreover, maximal reduction in Y(II) and qP was found at 100 μmol m−2 s−1 in hybrid P3939 (Table 1; Figure 1). Electron transport rate through PSII [ETR(II)] significantly increased as a function of light intensity in all three maize hybrids under normal or drought-stress conditions (Figure 1). In addition, drought stress reduced the ETR(II) at increasing light intensity greater than 100 μmol m−2 s−1, only in drought-tolerant and moderately tolerant maize hybrids 30Y87 and 31R88 (Figure 1). In drought-sensitive maize hybrid P3939, it remained almost unchanged at all actinic light intensities, except at 500 μmol m−2 s−1 where it decreased.

Nonphotochemical quenching (NPQ) increased with increase in light intensity in all three maize hybrids. Drought stress reduced the NPQ in maize hybrids 30Y87 and 31R88, whereas it increased the NPQ in maize hybrid P3939 at 300 μmol m^−2^ s^−1^ and beyond (Table 2; Figure 2). The fraction of regulated energy dissipation Y(NPQ) increased as a function of PAR in all three maize hybrids. Maize hybrids differed in Y(NPQ) under normal or drought-stress conditions. Development of NPQ as a function of PAR was more rapid in hybrid 31R88 than in other maize hybrids under drought stress. In addition, Y(NPQ) was significantly lower in hybrid P3939, particularly at the highest PAR 831 μmol m^−2^ s^−1^. Drought stress increased nonregulatory energy dissipation Y(NO) in hybrids 30Y87 and hybrid 31R88, whereas it remained almost unchanged in drought stressed plants of hybrid P3939 (Table 2; Figure 2).

#### 2.1.2. Measurement of Activity of PSI, ETR(I), Y(ND), Y(NA) and CEF

Electron transport through photosystem I ETR(I) and cyclic electron flow (CEF) were increased as a function of PAR in all three maize hybrids grown under normal or drought-stress conditions. Drought stress reduced both ETR(I) and CEF in maize hybrids 30Y87 and 31R88, while ETR(I) and CEF were strongly activated in P3939 (Figure 3).

Quantum yield of PSI [Y(I)] decreased as a function of PAR in all three maize hybrids under normal or drought-stress conditions. Drought stress increased the Y(I) at lower light intensities (i.e., less than 100 μmol m^−2^ s^−1^), and it reduced Y(I) at increasing actinic light intensities in hybrids 30Y87 and 31R88, whereas it was substantially higher in drought-stressed plants of hybrid P3939 (Table 3; Figure 4). Changes in Y(I) with change in actinic light intensity or drought stress can be related to either donor end limitation [Y(ND)] or acceptor end limitation [Y(NA)]. As a function of increasing actinic light intensities, Y(ND) increased and Y(NA) decreased in all three maize hybrids under normal or drought-stress conditions (Table 3; Figure 4). Drought stress reduced the Y(ND) in hybrid 31R88 and it did not change in hybrid 30Y87. In addition, it substantially increased the Y(ND) in drought-stressed plants of hybrid P3939. Drought stress also increased Y(NA) in hybrids 30Y87 and 31R88, particularly at 300 μmol m^−2^ s^−1^ and higher than this. In contrast, Y(NA) become almost zero in the P3939 maize hybrid under drought stress (Table 3; Figure 4).

### 2.2. Measurement of Leaf Water Potential and Yield of Maize

Leaf water potential (Ψ_L_) was significantly decreased (more negative values) in three maize hybrids due to the imposition of three cycles of drought (Table 4; Figure 5). However, the magnitude of reduction in Ψ_L_ was more pronounced in hybrid P3939 than that of the other two maize hybrids. Imposition of three cycles of drought caused a significant reduction in yield and yield attributes such as cob weight, kernel number per cob and 1000-seed weight in the three maize hybrids. A minimum reduction in cob weight (7%) and number of kernels per cob (56%) were observed in hybrid 30Y87 (Figure 5). While the maximum reduction (78%) in cob weight and number of kernels per cob (88%) was observed in P3939 in response to drought stress (Figure 5). Moreover, 1000-seed weight was also maximum in the 30Y87 maize hybrid, and the reverse was true for hybrid P3939 (Figure 5).

## 3. Discussion

Drought stress caused a reduction in growth and yield of all the three maize hybrids (Figure 5). Maize hybrids were ranked as drought tolerant (30Y87), moderately drought tolerant (31R88), and drought sensitive (P3939) based on biomass (data not shown) and yield, as shown in Figure 5E. The extent of reduction (40–90%) in maize yield depends on type of variety/hybrid, developmental stage at which drought stress is imposed and severity and duration of drought stress [4,26,27,28]. Severity of drought stress can be estimated from leaf water potential as it is an indicator of soil moisture content. While working with maize genotypes, Messmer, et al. [29] reported that leaf water potential of maize plants was positively correlated with soil moisture deficit, particularly in 30–70% soil moisture deficit range. Thus, leaf water potential is a potential indicator of drought stress. In the present study, at the beginning of the experiment in the fields, soil water contents in normal and drought-stressed soil were similar. Imposition of cyclic drought stress or withholding irrigation reduced the soil moisture and leaf water potential of the three maize hybrids (Figure 5). However, maximum reduction in leaf water potential was found in drought-sensitive hybrid P3939. Thus, plant water status is positively correlated with degree of drought tolerance. These results indicate that drought-tolerant maize hybrid 30Y87 had some drought tolerance mechanism, such as greater uptake of water through roots or lower loss of water through transpiration, or more retention of water through osmotic adjustment [19,30]. It is well-established that water stress causes an increase in xylem ABA that results in stomatal closure to maintain plant water status, and also to regulate influx of CO_2_ and photosynthetic rate [31]. It is proposed that in this study, cyclic drought stress reduced the photosynthesis in the three maize hybrids. Here, a question arises of whether impairment of photosynthesis under drought stress occurred either due to low leaf water potential-induced stomatal closure or due to photoinhibition of photosystems. Long-term stomatal closure to avoid water loss causes harmful effects such as photoinhibition of PSII due to generation of ROS, etc. Therefore, the relative contributions of photoprotection of photosynthetic machinery or antioxidant metabolism to reducing negative effects on photosynthesis under drought are of great importance.

Quantum yield of PSII is generally positively associated with photosynthetic rate in different crop plants [6,15,32,33]. Drought stress decreased the quantum yield of PSII, photochemical quenching and linear electron transport through PSII in maize hybrids 30Y87 and 318R88, which indicates photoinhibition of PSII occurred [19]. Since Y(II) is the output of qP or efficacy of absorbed energy by open PSII reaction centers as electron transport rate, drought-induced reduction in Y(II) and ETR(II) occurred at the expense of an increase in nonphotochemical quenching (NPQ). Increase in NPQ might have been either due to an increase in photoinhibition of PSII or photoprotective NPQ [24,34]. Our results showed that drought stress did not significantly affect Y(NPQ) as a function of actinic light intensity in all the three maize hybrids. However, drought stress increased Y(NO) in drought-tolerant maize hybrid 30Y87 and moderately tolerant hybrid 31R88, indicating that decline in Y(II) and ETR in the three maize hybrids was not associated with photoprotective NPQ, as has been observed in some of previous studies [17,19,21,24]. However, Y(NPQ) increased in drought-stressed plants of drought-sensitive hybrid P3939, along with a concomitant decrease in Y(NO). In addition, changes in Y(NPQ) and Y(NO) are associated with changes in Y(II) and qP. The results from this study indicated that Y(NO) was mainly associated with the reduced activity of PSII and increased fraction of closed PSII RCs in these maize hybrids. These results are in contrast to the findings of many studies, in which greater Y(NPQ) was found to be associated with better Y(II), overall photosynthetic capacity and growth of plants such, as in the mung bean [19], maize [35], and *Jatropha curcas* [21]. However, these results can be explained in view of the arguments of Murchie and Ruban [22], that the contribution of NPQ in photoprotection of PSII. Y(II) differs according to type of species or cultivars of same species. Changes in values of both Y(NPQ) and Y(II) and drought tolerance depend on specific processes for the utilization of a proportion of absorbed excitation energy in biochemical reactions for CO_2_ fixation. This has been further supported by the recent updates about the molecular mechanisms of NPQ development in different species [24,34].

The three maize hybrids had greater ETR(I) than ETR(II), which showed higher functional status of PSI than that of PSII under normal or drought-stress conditions. In addition, proportion of PSI-active centers in the three maize hybrids were higher under drought stress. This fortifies the idea that these additional PSI-active centers had a role in CEF to stimulate the proton-motive force that results in non-photochemical quenching (NPQ) to protect PSII [24,36]. In this study, drought stress slightly decreased CEF in 30Y87 but it did not change CEF in 31R88 under drought. This indicated that these two maize hybrids efficiently balanced the electron transport from PSII to PSI and its utilization in CO_2_ fixation. However, drought stress substantially increased CEF in drought-sensitive hybrid P3939. These results can be explained in view of arguments of Ruban and Wilson [24], that cyclic electron transport not only participates in development of proton-motive force for ATP synthesis and generation of Y(NPQ), but it also helps in avoiding over-reduction of PSI and generation of ROS under drought stress. In drought-sensitive hybrid P3939, a substantial increase in CEF is coupled with a slight increase in Y(NPQ) with no change in Y(II), suggesting a role of increased CEF in protecting PSII, as found in *Jatropha curcas* [21].

Changes in Y(I) and ETR(I) might have been due to either PSI donor end limitation or PSI acceptor end limitations. Our results showed that drought stress increased the Y(NA) and decreased the Y(ND) in maize hybrids 30Y87 and 31R88. In contrast, in drought-stress-sensitive hybrid P3939, Y(ND) substantially increased with almost zero Y(NA). The increase in donor-side limitation indicated greater electron demand at PSI than the transport of electron from PSII or lower electron transport from PSII to PSI due to PSII photoinhibition [32]. Y(ND) is also a measure of PSI donor-side limitations causing the non-photochemical energy dissipation in PSII [37]. From these results and these reports, it is suggested that drought stress reduced the biochemical processes that are downstream to PSII and PSI, such as CO_2_ fixation. The drought-tolerant and moderately drought tolerant maize hybrids 30Y87 and 31R88 downregulated activities of PSII, PSI and electron transport to avoid ROS generation and photoinhibition of photosystems. However, drought-sensitive maize hybrid P3939 was unable to downregulate PSII and electron transport from PSII to PSI. However, cyclic electron transport substantially increased in drought-stressed plants of drought-sensitive hybrid P3939, which resulted in an increase in Y(NPQ) at 500 and greater than 500 µmol m^−2^ s^−1^ actinic light, to protect photosystems. Recently, it has been observed that upon drought-stress-induced increase in Y(NA) or inhibition CO_2_ fixation, excessive electrons from PSI are shunted away to the water–water cycle to protect photoinhibition of photosystems [38]. However, the contribution of the water–water cycle was around 5% that of the linear electron flow regarding protection of photosystems from photoinhibition [38,39]. Thus, these three maize hybrids used a different combination of mechanisms to mitigate excess energy induced by drought, as has been observed earlier in different plant species such as mung bean [19], *Jatropha curcas*, *Ricinus communis* [21], and *Hordeum vulgare* [40]. However, drought-sensitive maize hybrid P3939 did not translate increased CEF and Y(NPQ) to increased yield, as drought stress substantially reduced the number of kernels per cob and cob weight in maize hybrid P3939 by 88% and 78%, respectively. However, downregulation of PSII and LEF in drought-tolerant hybrid 30Y87 might have produced enough ATP and NADPH to exactly match their consumption in the Calvin cycle, thus, translating it into better yield, i.e., 30Y87 had the maximum number of kernels per cob, cob weight and 1000-seed weight. So, kernel number not only characterizes itself as cob biomass accumulation, but it has also been associated with specific physiological and developmental processes.

## 4. Materials and Methods

### 4.1. Pot Experiment

In order to assess the genotypic variation in photosynthetic electron flux and photoprotective mechanism, an experiment was conducted at The Women University, Multan, Pakistan. In this pot experiment, three maize hybrids (30Y87, 31R88, P3939) were evaluated for photosynthetic acclimatory responses to three cycles of drought stress (Control, 3 drought cycles (D3)), arranged in a randomized complete block design with four replicates. Seeds of each maize hybrid were surface-sterilized with 5% sodium hypochlorite solution for 5 min and then rinsed with distilled water thrice. Seven surface-sterilized seeds of each maize hybrids were sown in plastic pots (24 cm Ø) filled with 12 kg garden soil. After sowing, the plastic pots were watered with tap water to field capacity to initiate seedlings’ germination. Then, one-week-old maize seedlings were thinned to four plants per pot with almost uniform size, and placed equidistantly. After two weeks, plants were subjected to drought stress by withholding watering up to the wilting stage, and then rehydrated, which was considered as one drought cycle. If 80% of the leaves of a plant were wilted, the plant was considered as wilted. If more than two thirds of the plants of a maize hybrids were wilted, the maize hybrid was considered as wilted. When plants of a maize hybrid begun wilting, droughted wilting plants and the corresponding control plants of that maize hybrid were rehydrated to field capacity. This was repeated thrice, or three drought cycles were imposed. The first drought cycle was slightly longer and occurred after 10–11 days, depending upon the genotype; the second and third cycles occurred after seven and five days, respectively. All measurements for physiological attributes were recorded after the completion of three cycles of drought stress, when the age of plants of three maize hybrids were five weeks.

#### 4.1.1. Measurement of PSII and PSI Activities as Rapid Light Curve Response

Activity of PSII and PSI was measured on the youngest but fully developed leaf, usually the third leaf from the top, in the three maize hybrids using rapid light curve response with DualPAM-100 (Walz, Germany) following manufacturer’s instructions. The activities of both PSII and PSI were measured simultaneously by setting the range of light intensities. Before the measurements, leaves were dark-adapted for 30 min by wrapping leaves with aluminum foil. Fluorescence attributes were recorded after 40 s of exposure to PPFD (11, 18, 27, 58, 100, 131, 221, 344, 536, and 830 μmolm^−2^s^−1^). Actinic light of all light intensities was applied for 40 s, and saturation pulse of 600 μmolm^−2^s^−1^ with wavelength of 660 nm was applied for 0.8 s. Complementary quantum yields of PS(II) and PS(I) were automatically recorded in DualPAM software. Description of measured attributes of PS(I) and PS(II) obtained from LCR are described in Table 5.

#### 4.1.2. Measurement of Leaf Water Potential

The youngest fully developed, usually the third leaf from the top, was selected and excised from each replicate of all treatments to measure leaf water potential. Measurements for leaf water potential were made at 8:00 am using Scholander type apparatus (Chas W. Cook and Sons, Birmingham, UK).

### 4.2. Field Experiment

The field study was conducted in the Research Field area of the Women University, Multan Pakistan, to assess the yield potential in maize under drought stress. The field soil, where the experiment was conducted, is loam in texture. During the experimentation, the average maximum and minimum temperatures were 36.67 °C ± 21.5 °C and the rainfall ranged from 19.5 to 12.3 mm during the growing season of maize (March–June, 2018). Before the start of experimentation, soil samples were collected at a depth of 30 cm and analyzed for soil pH, EC, organic matter, soil texture and soil saturation percentage. The soil of the research field was silty clay loam with 40% soil saturation percentage. The soil pH was measured by making a soil suspension with water in 1:2.5 ratio, while soil EC was measured by making a soil suspension of 1:5. The soil pH_(1:2.5)_ was 7.8 and EC_(1:5)_ was 0.34 mS/cm. The soil organic content was 1.32%. In this experiment, seeds of maize hybrids (30Y87, 31R88 and P3939) were sown on the ridges. The experimental layout was a completely randomized factorial design with three replications. The plot area was 12 × 12 ft^2^ (144 ft^2^). Each plot area consisted of 6 rows, 12 ft (3.6 m) long. Each row was spaced with 2 ft (0.60 m), while spacing between plants was 0.3 m. Plants of each maize hybrid were irrigated as normal, termed control, and under water stress. Water stress was applied at the vegetative stage of maize hybrids by withholding irrigation. Watering was discontinued in water-deficit treatments up to wilting and leaf rolling stage. However, it was ensured that under the drought-stress condition soil water content would remain higher than that which would create a permanent wilting stage, or greater than 13% soil water content. The well-watered plants continued to irrigate weekly, up to field capacity and until physiological maturity. Sowing was conducted manually, and 2 seeds per hill were placed at a 20 cm plant distance. Thinning of the plants was conducted manually, to maintain one plant per hill.

Nitrogen was applied as urea (NH_2_-CO-NH_2_ (46% N) at the rate of 100 kg/ha, DAP at 60 kg/ha, and 50 kg/ha potassium sulphate applied at sowing, at 60 cm height and in the tasseling stage. Maize cultivars were kept free of weeds by hoeing to avoid weed–crop competition. The plot area (12×12 ft^2^) of each treatment was harvested and 10 subsamples of each maize hybrid were taken for the analysis of various yield attributes. The following yield components were measured according to standard procedures.

#### Yield Attributes

Weight of cob (g): 10 cobs selected randomly from plot area (12 × 12 ft^2^) of each maize hybrid and weighed by means of electric balance.

Number of kernels per cob: Number of kernels of 10 cobs from each plot area were counted and averages were taken.

1000-kernel weight (g): The data for this attribute was recorded at random from the grain lot of each plot area and weighed by an electric balance.

### 4.3. Statistical Analysis

The data obtained were subjected to three-way analysis of variance using statistical computer package CoStat v. 6.3 (Cohort, Davis, CA, USA). Means were compared using LSD following Snedecor and Cochran (1980). The data for rapid light curve response were drawn as a scatter plot with the lines and marker option using MS-Excel 2010. The data for leaf water potential and yield attributes were drawn as bar charts. The ANOVA tables and charts were presented as composite tables and composite graphs.

## 5. Conclusions

The adverse effects of drought on growth and yield of maize hybrids were variable. Genetic variability in maize hybrids was associated with maintenance of plant water status. The drought-tolerant maize hybrid 30Y87 downregulated PSII activity and electron transport from PSII to PSI to a sufficient extent, which was consistent with the need for downstream biochemical processes, and resulted in lower yield losses under drought. However, the increase in cyclic electron flow and the photoprotection component of NPQ in the drought-sensitive maize hybrid P3939 did not improve the yield. Thus, we confirm that the photoprotection of PSII does not play the only role in drought tolerance of maize. Such a mechanism requires coordination with other physiological processes for the induction of drought tolerance.

## Figures and Tables

**Figure 1 ijms-22-13490-f001:**
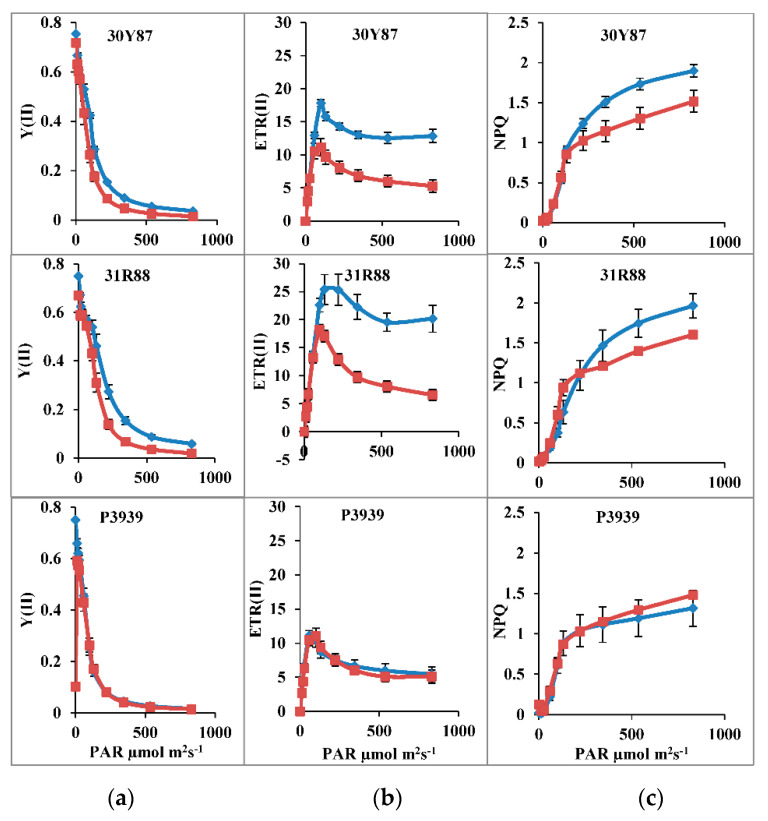
Analysis of PSII photochemistry of three maize hybrids when two-week-old plants were subjected to three cycles of drought stress. (**a**) Effective quantum yield of electron transport at PSII [Y(II)]; (**b**) Electron transport rate through photosystem II [ETR(II)]; (**c**) Nonphotochemical quenching (NPQ). Blue lines are control, red lines with markers are water stress. Bars represent the calculated standard error (*n* = 4).

**Figure 2 ijms-22-13490-f002:**
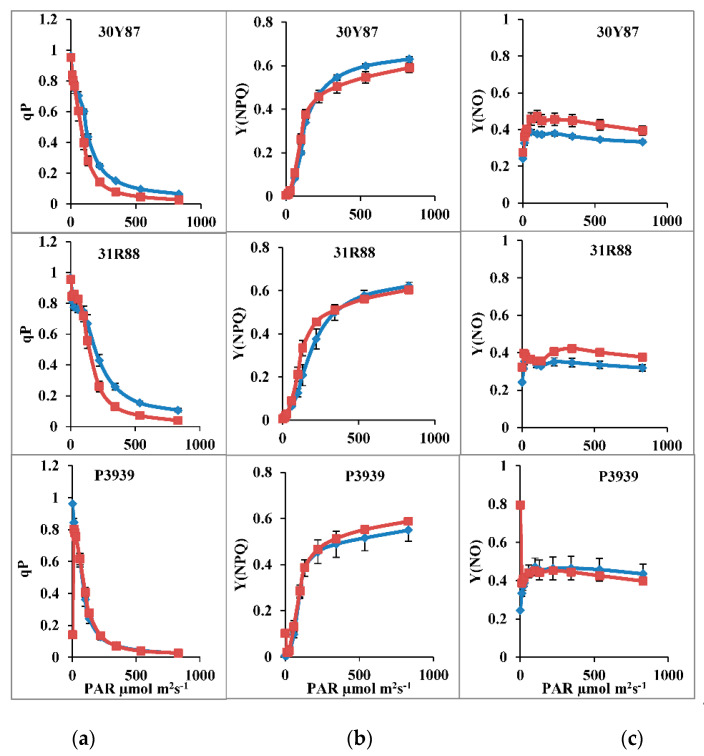
Changes in photochemistry qP, Y(NPQ), and Y(NO) of three maize hybrids, when two-week-old plants were subjected to three cycles of drought stress. (**a**) photochemical quenching (qP); (**b**) Fraction of energy dissipated as heat via the regulated nonphotochemical quenching mechanism [Y(NPQ)]; (**c**) Fraction of energy that is passively dissipated as heat and fluorescence [Y(NO)]. Blue lines are control, red lines with markers are water stress. Bars represent the calculated standard error (*n* = 4).

**Figure 3 ijms-22-13490-f003:**
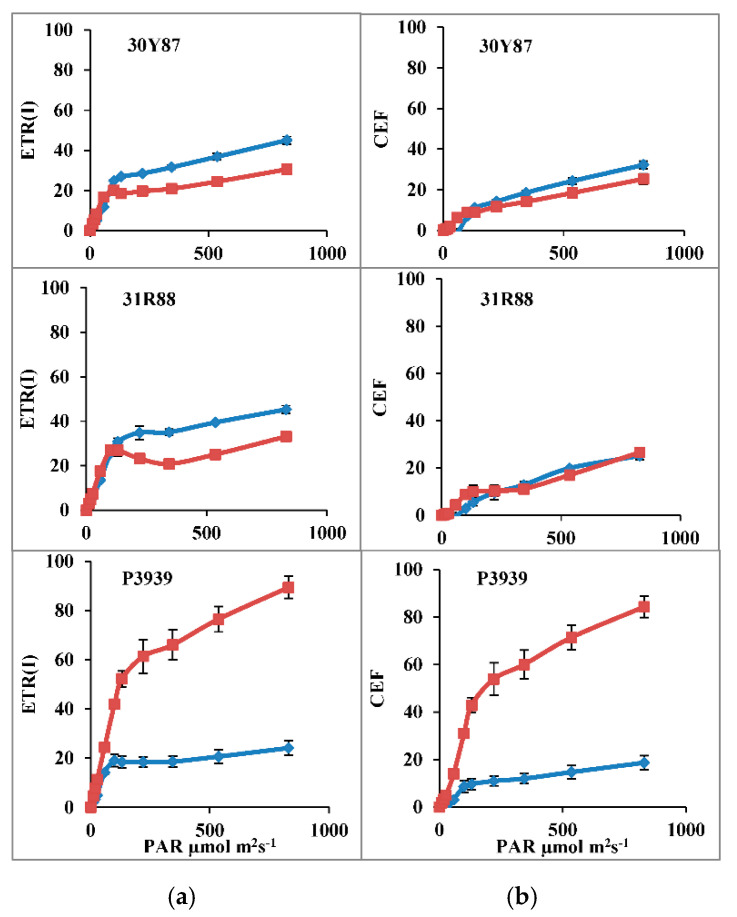
Changes in ETR(I) and CEF of three maize hybrids when two-week-old plants were subjected to three cycles of drought stress. (**a**) Electron transport rate in PS(I) [ETR(I)] and (**b**) Cyclic electron flow [CEF]. Blue lines are control, red lines with markers are water stress. Bars represent the calculated standard error (*n* = 4).

**Figure 4 ijms-22-13490-f004:**
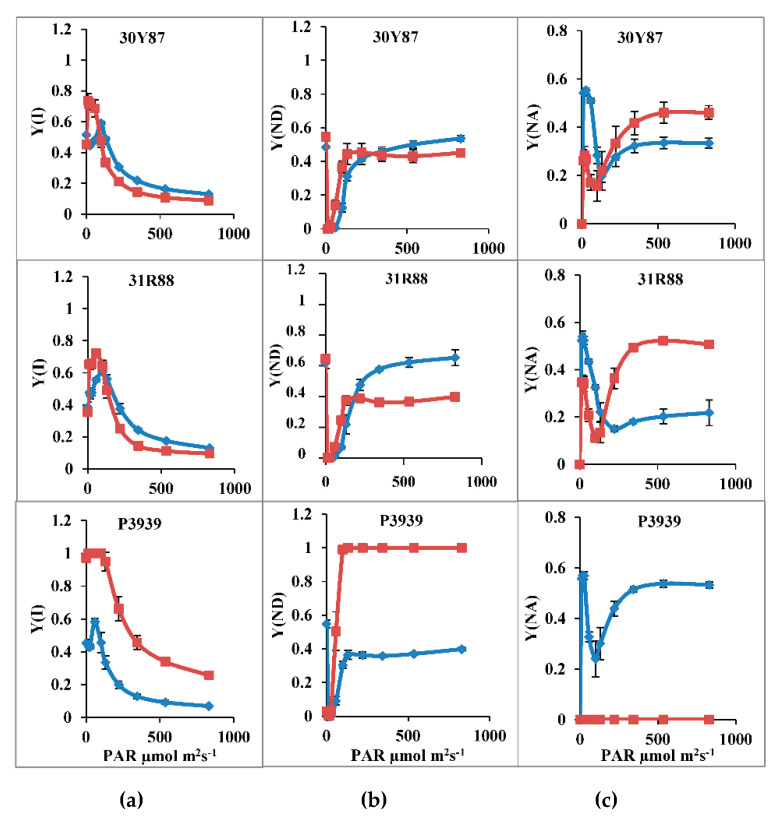
Activity of PSI of three maize hybrids when two-week-old plants were subjected to three cycles of drought stress. (**a**) Effective quantum yield of photosystem I [Y(I)] (**b**) fraction of overall P700 that is oxidized in a given state [Y(ND)] (**c**) Fraction of overall P700 that cannot be oxidized in a given state [Y(NA)]. Blue lines are control, red lines with markers are water stress. Bars represent the calculated standard error (*n* = 4).

**Figure 5 ijms-22-13490-f005:**
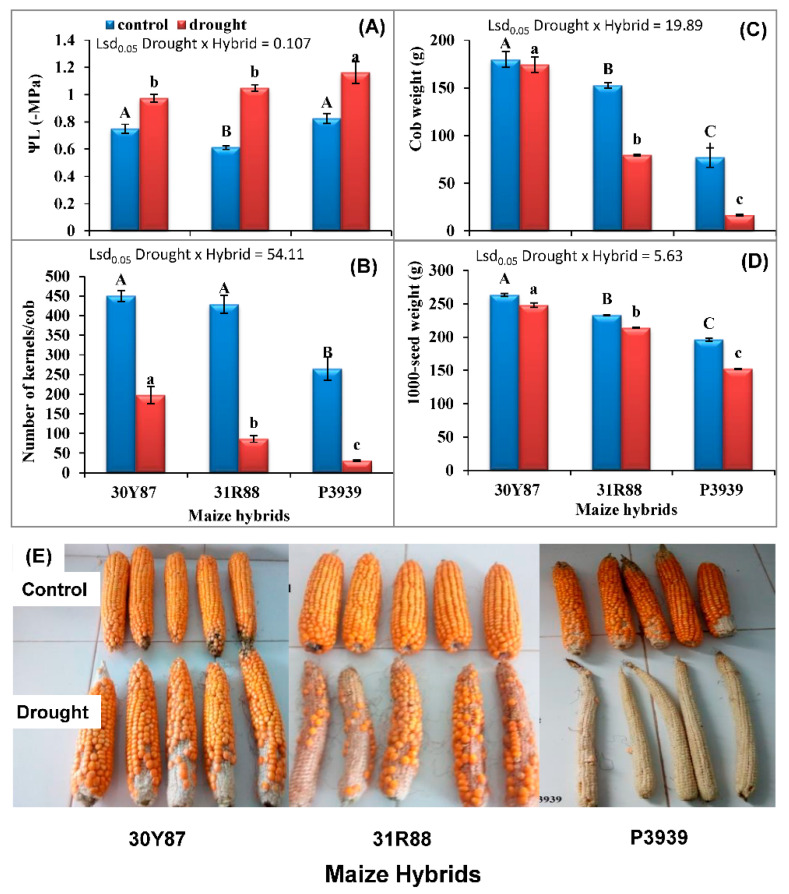
Leaf water potential (Ψ_L_) (**A**), number of kernels/cob (**B**), cob weight (**C**), and 1000-seed weight (**D**) of three maize hybrids when two-week-old plants were subjected to three cycles of drought stress. Blue bars are control, red bars with markers are water stress. Bars represent the calculated standard error (*n* = 4 for leaf water potential and *n* = 12 for yield attributes from field experiment). (**E**) Cobs of three maize hybrids grown under normal or drought conditions reflecting maize hybrids yield.

**Table 1 ijms-22-13490-t001:** Mean square values from analysis of variance (ANOVA) for Y(II), ETR(II), and qP of three maize (*Zea mays* L.) hybrids differing in drought tolerance when two-week-old plants were subjected to three cycles of drought stress.

Scheme	df	Y(II)	ETR(II)	qP
**Treatment**	1	0.304 ***	718.7 ***	0.229 ***
**PAR**	10	1.500 ***	566.6 ***	2.481 ***
**Maize hybrids.**	2	0.234 ***	725.4 ***	0.631 ***
**Treatment * PAR**	10	0.0261 ***	52.50 ***	0.040 ***
**Treatment * hybrids.**	2	0.001 ns	178.7 ***	0.006 *
**PAR * hybrids.**	20	0.028 ***	52.96 ***	0.054 ***
**Treatment * PAR * hybrids.**	20	0.028 ***	18.77 ***	0.053 ***
**Error**	198	9.635	2.930	0.001 ***
**Total**	263			

ns = non-significant; *, **,*** significant at 0.05, 0.01 and 0.001 probability levels, respectively.

**Table 2 ijms-22-13490-t002:** Mean square values from analysis of variance (ANOVA) for Y(NPQ), NPQ and Y(NO) of three maize (*Zea mays* L.) hybrids differing in drought tolerance when two-week-old plants were subjected to three cycles of drought stress.

Source of Variation	df	Y(NPQ)	NPQ	Y(NO)
**Treatment**	1	1.244 ***	1.055 ***	0.005 ns
**PAR**	10	2.042 ***	5.173 ***	0.090 ***
**Maize hybrids.**	2	1.112 ***	1.174 ***	0.030 ***
**Treatment * PAR**	10	0.065 ***	0.820 ***	0.033 ***
**Treatment * hybrids.**	2	1.025 ***	0.189 ***	0.246 ***
**PAR * hybrids.**	20	0.062 ***	0.777 ***	0.046 ***
**Treatment * PAR * hybrids.**	20	0.073 ***	0.332 ***	0.020 ***
**Error**	198	0.001	0.023	0.001
**Total**		263		

ns = non-significant; *, **, *** significant at 0.05, 0.01 and 0.001 probability levels, respectively.

**Table 3 ijms-22-13490-t003:** Mean square values from analysis of variance (ANOVA) for Y(I), ETR(I), Y(ND) and Y(NA) of three maize (*Zea mays* L.) hybrids differing in drought tolerance when two-week-old plants were subjected to three cycles of drought stress.

Source of Variation	df	Y(I)	ETR(I)	Y(ND)	Y(NA)
**Treatment**	1	2.027 ***	2357.4 ***	0.882 ***	1.769 ***
**PAR**	10	1.031 ***	5728.9 ***	1.383 ***	0.324 ***
**Maize hybrids.**	2	0.616 ***	1931.1 ***	0.639 ***	0.303 ***
**Treatment * PAR**	10	0.092 ***	83.53 ***	0.131 ***	0.122 ***
**Treatment * hybrids.**	2	1.304 ***	6970.2 ***	0.878 ***	1.099 ***
**PAR * hybrids.**	20	0.017 ***	141.9 ***	0.114 ***	0.008 ***
**Treatment * PAR * hybrids.**	20	0.020 ***	617.7 ***	0.127 ***	0.059 ***
**Error**	198	0.002	11.18	0.002	0.002
**Total**	263				

ns = non-significant; *,**,*** significant at 0.05,0.01 and 0.001 probability levels, respectively.

**Table 4 ijms-22-13490-t004:** Significance level from analysis of variance observed for leaf water potential (-MPa), 1000-seed weight, cob weight and number of kernels/cob of three maize (*Zea mays* L.) hybrids when two-week-old plants were subjected to three cycles of drought stress.

Source of Variation	df	Ψ_L_	1000-Seed Weight	df	Cob Weight	Kernel No./Cob
**Treatment**	1	0.666 ***	4024.08 ***	1	32155 ***	1121760 ***
**Maize hybrids**	2	0.059 ***	13426.5 ***	2	84941 ***	150059 ***
**Treatment * Maize hybrids**	2	0.022 *	478.2 ***	2	6508 ***	19006.6 **
**Error**	18	0.005	14.38	54	492.5	3641.6
**Total**	23			59		

ns = non-significant; *,**,*** significant at 0.05,0.01 and 0.001 probability levels, respectively.

**Table 5 ijms-22-13490-t005:** Chlorophyll fluorescence measurements using Dual-PAM100.

Parameters	Physiological Interplay	Calculated Formula
Y(II)	Effective quantum yield of PSII	Φ_PSII_ = (F_M_′ − F_S_′/F_M_′) = ΔF/F_M_
ETR(II)	PSII electron transport rate	ETR = Y(II) × PAR × 0.42
NPQ	Nonphotochemical quenching of Fm	(Fm − Fm′)/Fm′
qP	Photochemical quenching based on ‘’puddle’’ model	(Fm′ − Fs′)/(Fm′ − Fo′)
Y(NPQ)	Quantum yield of pH-dependent energy dissipation in PSII	Φ_NPQ_ = 1 − Φ_PSII_ − 1/[NPQ + 1 + qL(F_M_/F_O_ − 1)] Kramer et al., 2004)
Y(NO)	Quantum yield of nonregulated dissipation of energy in PSI	Φ_NO_ = 1/[NPQ + 1 + qL × (F_M_/F_O_ − 1)] Kramer et al., 2004)
Y(I)	Effective quantum yield of PSI photochemistry	(Pm′ − P/Pm)
ETR(I)	PSI electron transport rate	ETR = Y(I) × PAR × 0.42
Y(ND)	Fraction of total P700 that is oxidized due to donor-side limitation	P/Pm
Y(NA)	Fraction of total P700 that cannot be oxidized due to lack of acceptors	Pm − Pm′/Pm
Y(CEF)	Cyclic electron flow estimated by Y(I) − Y(II)	Y(I) − Y(II)

## Data Availability

Data may be provided upon request.

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
