# Peer review of "Is Photoprotection of PSII One of the Key Mechanisms for Drought Tolerance in Maize?"

_ijms, 2021, doi:10.3390/ijms222413490_

Round 1
Reviewer 1 Report
Review of IJMS-1426189
Is Photoprotection of PSII one of the Key Mechanisms for Drought Tolerance in Maize?
Nahidah Bashir, Habib-ur-RehmanAthar, Hazem M. Kalaji, Jacek Wróbel,Seema Mahmood, ZafarUllah Zafar and Muhammad Ashraf
The authors wished to study the basis for drought tolerance in maize. They therefore selected 3 maize hybrids known to vary in drought tolerance, and subjected them to three cycles of drought stress to the point of wilting. They then measured their water potentials and a number of parameters related to the light reactions of photosynthesis. Overall, they identified several significant differences in these parameters in response to drought between the drought sensitive and drought-resistant hybrids. They also measured the overall yield of the three hybrids under control conditions and after drought stress, and identified significant differences.
Overall, their results are interesting. They seem to have been performed competently, with suitable instruments and adequate replication. They are therefore worth sharing with the plant community after correcting the issues noted below.
My biggest concern was that they didn’t single out any targets for improving crop yield under drought stress. How can these data help maize breeders identify superior plants for their breeding programs? Since much of their data was based on measurements that can easily be obtained in the field with relatively inexpensive equipment, what parameters should breeders be looking for when assessing their plants growing in the field?
I also found it difficult to assess how they answered the question they asked in their title “Is photoprotection of PSII one of the key mechanisms for drought tolerance in maize?” They have clearly identified significant differences between these three hybrids in the ways that their light reactions respond to drought, so I would like to see them answer this question directly in their discussion. Again, I would like to see them address how this information could be used to improve drought tolerance in crop plants, since as the authors correctly note, this is a major problem that will grow every year.
Other points
Figures 3 and 4 were cut off in the pdf I was provided so I could not see the full figures.
Captions to figures 1, 2, 3, 4 and 5 should indicate what the blue and orange lines represent. These figures should also show error bars at each light intensity, and the captions should indicate the numbers of replicates and whether the errors are ± standard deviation or ± standard error.
Overall the English is good, but there are numerous mistakes and some make the meaning hard to understand. I therefore recommend editing by a native English speaker.
Since the lines aren’t numbered it is difficult to cite specific instances, but here are some examples.
The last two lines on page 1 are grammatically incorrect and hard to understand.
These sentences near the top of page 2 are also grammatically incorrect and hard to understand. “Stomatal closure leads to the reduction in CO2 assimila-tion that ultimately declines the enzymatic activity of the photosynthesis under drought conditions [3, 4]. As a result of perturbation in photosynthesis with maximum irradiance under drought conditions can produce excess excited energy index, results in accumula-tion of reactive oxygen species (ROS) that put down the D1 protein synthesis and cause the photo damage to photosystem II (PSII) [5, 6].”
These sentences near the bottom of page 2 are also grammatically incorrect and hard to understand. “However, controlled oxidative damage is one of the slot of photosynthetic system can protect the rest of machinery against the photodamage. As D1 protein of PSII, considered as “suicide protein” scarified to save the rest of PSII ma-chinery against oxidative damage under excess light [12]. Moreover, when photoinhibi-tion has set up, the PSII reaction centers is concurrently repaired by exclusion, synthesis and replacements of damaged D1 protein. For example in an experiment on Lotus, PSII activity drastically decreased under combined drought-heat stress and stimulatingly this correlate to degradation of D2 protein [13]. This reveals that stress responsive capability to oxidative damage and adaptation of photosynthetic machinery are the physiological characteristics manifest the tolerance of plant species against environmental stresses. This reveals that stress responsive capability to oxidative damage and adaptation of photosynthetic machinery are the physiological characteristics manifest the tolerance of plant species against environmental stresses.”
Top of page 4: I do not understand “and the relative humidity range from 19.5 to 12.3 mm during the growing season of maize (March-Jun, 2018).” Are you referring to the amount of rainfall during this period?
Top of page 11: this sentence is both very important and hard to understand “Our results clearly indicated that reduction in ETR(II),Y(II),and (qP) in 30Y87 and 31R88 might be involve to avoid the over oxidization-reduction of proton circuit along PSII and PSI which might be helpful to regulate the PS(II) activity against the photo-dam-aging effects due to drought.”
Author Response
Dear Sir
Thank you very much for your positive criticism. This helped a lot in improving the quality of the MS. Please find attached responses point by point.
Hope you will find the MS suitable for publication in IJMS now.
Best Regards

Reviewer 2 Report
The presented manuscript addresses the current issue of the impact of water deficit on maize plants. Classical physiological methods, especially fluorescence methods, were used as suitable parameters for determining the resistance of maize plants. This is a current issue, as water deficit is one of the most important stressors, which affects not only the metabolism of plants, but also their subsequent production. The manuscript is written carefully, chronologically. English is also at a good level. I would add values to the abstract that would demonstrate the differences found. In the methodological part I miss the exact definition of water stressor induction in field and laboratory experiments. How was the water deficit of the soil determined? In field experiments, it is necessary to supplement the characteristics of the soil - nutrients, pH, humus content. The results are descriptive. It would be appropriate to add values or differences to the text. I recommend adding the values of standard deviations to Tables 2 to 5. In graphs 1-4, add descriptions of the axes.
Author Response
Dear Sir
Thank you very much for your positive and constructive criticism. The MS has been revised in view of your suggestions and incorporated all suggested points in various sections of the MS. Hope you will find the MS suitable for its publication in IJMS. Please find attached file.
Best Regards
Nahida Bashir

Round 2
Reviewer 1 Report
Review of revised IJMS-1426189
Is Photoprotection of PSII one of the Key Mechanisms for Drought Tolerance in Maize?
Nahidah Bashir, Habib-ur-RehmanAthar, Hazem M. Kalaji, Jacek Wróbel,Seema Mahmood, ZafarUllah Zafar and Muhammad Ashraf
The authors wished to study the basis for drought tolerance in maize. They therefore selected 3 maize hybrids known to vary in drought tolerance, and subjected them to three cycles of drought stress to the point of wilting. They then measured their water potentials and a number of parameters related to the light reactions of photosynthesis. Overall, they identified several significant differences in these parameters in response to drought between the drought sensitive and drought-resistant hybrids. They also measured the overall yield of the three hybrids under control conditions and after drought stress, and identified significant differences.
Overall, their results are interesting. They seem to have been performed competently, with suitable instruments and adequate replication (although the numbers of replicates need to be indicated in the figure and table captions, as noted below). They are therefore worth sharing with the plant community after correcting the issues noted below.
The authors have addressed many of the issues raised in my review of their initial submission, but there are still many points that should be addressed
It is not clear how old the plants were at the time the measurements were taken. From lines 308-322 it seems that plants were 2 weeks old when water was first withheld, but how long did it take to go through 3 rounds of drought to the wilting stage? Were all three hybrid lines sampled at the same time? Did it take all of them the same time to reach the wilting stage? Titles to all figures and tables should indicate the actual ages of the plants and that they had been subjected to three drought cycles starting when they were 2 weeks old.
Captions to all figures and tables are improved, but still need to indicate the numbers of replicates for each treatment and each hybrid.
Table 1 should come after Figure 1 since it is showing the statistical analysis of some of the data presented in Figure 1.
Overall the English is improved, but there are still numerous mistakes and some make the meaning hard to understand. Many involve problems with singular and plural or use of pronouns and don’t affect the meaning, but others require correction.
Lines 19-20: Please change to “… three maize hybrids were subjected to three drought cycles, then the activities of photosystem II (PSII) and photosystem I (PSI) were measured.”
Lines 45-53 are hard to understand and should be rewritten for clarity.
Lines 77, 91 and 311: Please change “acclamatory” to “acclimatory”
Lines 98-101: Please indicate that these parameters decreased in response to drought stress.
Line 209: It seems that the actual yield data are presented in figure 5. Figure 6 should be just another pane in figure 5, since it does not present any numerical data but shows the appearance of the cobs measured to obtain the data presented in figure 5.
Lines 215-217: You seem to indicate that soil moisture data is presented in figure 5, but all I see presented is leaf water potential and various yield parameters.
Lines 226-232 are hard to understand and should be rewritten for clarity.
Lines 248-251 are hard to understand and should be rewritten for clarity.
Lines 287-290 are hard to understand and should be rewritten for clarity.
Line 328: “warping” should be “wrapping”
Lines 358-362 are hard to understand and should be rewritten for clarity.
Author Response
The Editor in Chief
International Journal of Molecular Sciences (IJMS)
Subject: Submission of revised version of the MS, “Is Photoprotection of PSII one of the Key Mechanisms for Drought Tolerance in Maize?”
Dear Sir
With reference to comments raised by the reviewers on our MS, thank you very much for giving sufficient time to revise the MS. The MS has been revised keeping in view the comments raised by the reviewers. We addressed all comments and made revisions in all sections. I would like to submit revised version of the MS along with responses point by point.
Academic Editor
All Figures require some editing: insets are displaced and there is no indication in the legend on what the colors mean. Also please indicate what the X-axis represents. Figure 5 requires posthoc test.
Response: The Figure 5 has been modified now. After doing posthoc analysis, means were compared with LSD0.05 and letter were inserted as suggested by the Academic Editor. In all figure captions, the meaning of color has been explained now. X-axis in this figure has also explained now.
Reviewer 1.
- (x) Minor editing of English language and style required
Are the methods adequately described? |
( ) |
(x) |
( ) |
( ) |
Are the results clearly presented? |
( ) |
(x) |
( ) |
( ) |
Are the conclusions supported by the results? |
(x ) |
( ) |
( ) |
( ) |
Response: Thanks you very much for your constructive comments on our MS. The M&M and Results section has been revised in track changes as suggested by the reviewer. The MS has been critically read by the senior authors now.
- The authors have addressed many of the issues raised in my review of their initial submission, but there are still many points that should be addressed. It is not clear how old the plants were at the time the measurements were taken. From lines 308-322 it seems that plants were 2 weeks old when water was first withheld, but how long did it take to go through 3 rounds of drought to the wilting stage? Were all three hybrid lines sampled at the same time? Did it take all of them the same time to reach the wilting stage?
Response: The description of experiment has been explained now in M&M section as suggested by the reviewer.
- Titles to all figures and tables should indicate the actual ages of the plants and that they had been subjected to three drought cycles starting when they were 2 weeks old.
Response: The Figure captions and Table captions have been modified as suggested by the reviewer.
- Captions to all figures and tables are improved, but still need to indicate the numbers of replicates for each treatment and each hybrid.
Response: Number of replicates has been mentioned in all figures and Tables as suggested by the reviewer.
- Table 1 should come after Figure 1 since it is showing the statistical analysis of some of the data presented in Figure 1.
Response: The position of Table 1 has been changed as suggested by the reviewer.
- Overall the English is improved, but there are still numerous mistakes and some make the meaning hard to understand. Many involve problems with singular and plural or use of pronouns and don’t affect the meaning, but others require correction.
Response: The MS has been read critically and improved the language where required as suggested by the reviewer.
- Lines 19-20: Please change to “… three maize hybrids were subjected to three drought cycles, then the activities of photosystem II (PSII) and photosystem I (PSI) were measured.”
Response: This sentence has been rectified as suggested by the worthy reviewer.
- Lines 45-53 are hard to understand and should be rewritten for clarity.
Response: The suggested sentence has been modified as, “Since plant growth and yield is mainly depends on plant photosynthetic activity, several researchers are of view that selection of cultivars/varieties based on photosynthetic traits may help in developing high yielding and stress tolerant crop cultivars”
- Lines 77, 91 and 311: Please change “acclamatory” to “acclimatory”
Response: The word acclamatory has been replaced with acclamatory at suggested points in the text of the MS as pointed out by the reviewer
- Lines 98-101: Please indicate that these parameters decreased in response to drought stress.
Response: The description of results of these parameters has been changed as suggested. Changes in Y(II), qP, ETR(II) due to drought stress occurred at light intensities at or greater than 100 umol m-2 s-1. This has been described in the text as well.
- Line 209: It seems that the actual yield data are presented in figure 5. Figure 6 should be just another pane in figure 5, since it does not present any numerical data but shows the appearance of the cobs measured to obtain the data presented in figure 5.
Response: The Figure 6 has been merged with Figure 5 as suggested by the reviewer.
- Lines 215-217: You seem to indicate that soil moisture data is presented in figure 5, but all I see presented is leaf water potential and various yield parameters.
Response: We are sorry for such an ambiguous sentence. The sentence has been modified now. Leaf water potential is an indicator of soil moisture deficit or water stress that was experienced by plants. A reference has also been cited to support this argument.
- Lines 226-232 are hard to understand and should be rewritten for clarity.
Response: This sentence has been modified as suggested by the reviewer.
- Lines 248-251 are hard to understand and should be rewritten for clarity.
Response: The argument in Discussion section has been changed as highlighted by the reviewer.
- Lines 287-290 are hard to understand and should be rewritten for clarity.
Response: These sentences has been re-written to make it more clearer as suggested by the reviewer.
- Line 328: “warping” should be “wrapping”
Response: This word has be corrected as suggested.
- Lines 358-362 are hard to understand and should be rewritten for clarity.
Response: These lines in M&M section has been modified in track changes as suggested by the reviewer.
Hope you will find the revised version of the MS suitable for its publication in your valuable Journal IJMS.
However, still if you find some errors in it, please don’t hesitate to contact us for its rectification.
Best Regards
Nahidah Bashir
Reviewer 2 Report
The authors corrected the manuscript according to the reviewer's comments. After studying the colored manuscript and explaining the authors, it can be stated that its quality has improved after the changes have been incorporated. I agree with the above changes and accept them.
Author Response
Reviewer 2.
- The authors corrected the manuscript according to the reviewer's comments.After studying the colored manuscript and explaining the authors, it can be stated that its quality has improved after the changes have been incorporated. I agree with the above changes and accept them.
Response: Thank you very much for your valuable comments on our MS.
